# Do Different Types of Microphones Affect Listening Effort in Cochlear Implant Recipients? A Pupillometry Study

**DOI:** 10.3390/jcm13041134

**Published:** 2024-02-17

**Authors:** Sara Ghiselli, Erica Pizzol, Vincenzo Vincenti, Enrico Fabrizi, Daria Salsi, Domenico Cuda

**Affiliations:** 1Department of Otolaryngology, AUSL Piacenza, 29121 Piacenza, Italy; e.pizzol@ausl.pc.it (E.P.); d.salsi@ausl.pc.it (D.S.); d.cuda@ausl.pc.it (D.C.); 2Department of Otolaryngology and Otoneurosurgery, University of Parma, 43126 Parma, Italy; vincenzo.vincenti@unipr.it; 3Department of Economics and Social Sciences, Università Cattolica del S. Cuore, 29121 Piacenza, Italy; enrico.fabrizi@unicatt.it; 4Department of Medicine and Surgery, University of Parma, 43126 Parma, Italy

**Keywords:** cochlear implant, hearing stimulation, signal processing, listening effort, pupillometry

## Abstract

Background: It is known that subjects with a cochlear implant (CI) need to exert more listening effort to achieve adequate speech recognition compared to normal hearing subjects. One tool for assessing listening effort is pupillometry. The aim of this study is to evaluate the effectiveness of adaptive directional microphones in reducing listening effort for CI recipients. Methods: We evaluated listening in noise and listening effort degree (by pupillometry) in eight bimodal subjects with three types of CI microphones and in three sound configurations. Results: We found a correlation only between sound configurations and listening in noise score (*p*-value 0.0095). The evaluation of the microphone types shows worse scores in listening in noise with Opti Omni (+3.15 dB SNR) microphone than with Split Dir (+1.89 dB SNR) and Speech Omni (+1.43 dB SNR). No correlation was found between microphones and sound configurations and within the pupillometric data. Conclusions: Different types of microphones have different effects on the listening of CI patients. The difference in the orientation of the sound source is a factor that has an impact on the listening effort results. However, the pupillometry measurements do not significantly correlate with the different microphone types.

## 1. Introduction

Cochlear implantation is a standard of care for individuals with severe-to-profound hearing loss for whom conventional hearing aids are insufficient. Most cochlear implant (CI) recipients show good outcomes in quiet conditions [1], but do not perform well in speech understanding in noise. This is partly due to the loss of spectral and temporal resolution and the narrow electrical dynamic range of the CI device [2].

Advanced signal processing strategies can be beneficial in this specific area. The auditory results of CI subjects can be improved by applying technologies that increase the signal-to-noise ratio (SNR) [3]. Currently, the most common technology used to improve listening in noise is the use of those types of microphones that, depending on the location of the sound source, direct their pickup towards the listener. Dual-microphone beamformers are often used to improve the directionality of sound. In contrast, omnidirectional microphones are equally sensitive to signals coming from all directions [4,5].

While these signal strategies certainly lead to improved listening in noise, it has been reported that CI subjects need to exert more effort to achieve adequate sound and speech recognition compared to subjects with normal hearing [6]. This effort, known as listening effort, refers to the amount of processing resources allocated to a particular auditory task when task demands are high [7].

Recent studies have investigated listening effort using a variety of tools, including pupillometry. Pupillometry is based on the evaluation of changes in the pupil size during the presentation of sensory stimuli. An increase in task complexity implies an increase in pupil diameter. Listening to acoustic stimuli in a difficult environment involves an increase in listening effort, which is measured as an increase in pupil diameter [8,9]. However, few studies have measured the listening effort associated with the use of implantable hearing devices [10]. Other studies have demonstrated the relationship between pupillometry and hearing in noise. Ohlenforst et al. [11] determine the relationship between pupil dilation and SNR, and Gawecki et al. [12] assess listening effort in a group of subjects implanted with BAHS Ponto in noise.

The interest of pupillometry is that it can be applied for the development of new clinically relevant objective measures of speech perception in noise in CI users.

The aim of this study is to evaluate the effectiveness of adaptive directional microphones in reducing listening effort for CI recipients. We will also assess whether there is a correlation between hearing performance and listening effort as assessed by pupillometry.

## 2. Materials and Methods

### 2.1. Study Design

This is a monocentric, prospective, non-randomized, observational study conducted at the clinic of the otorhinolaryngology department at “Guglielmo da Saliceto” Hospital in Piacenza, Italy.

This study was approved by the Ethics Committee of the Area Vasta Emilia Nord under number 2020/0044523 on 21 April 2020 and was conducted according to Good Clinical Practice regulations and the Helsinki Declaration.

All subjects signed the Patient Informed Consent Form before the first assessment.

The participants in this study are bimodal patients and they use a CI in the worst hearing side and a hearing aid (HA) in the contralateral side. Everyone had been using the CI for 24 months.

### 2.2. Participants

Patients who had been implanted with the Neuro CI (Oticon Medical AB, Askim, Sweden) in an ear side and with a contralateral HA were included. Participants had to be over 18 years of age, use the Neuro 2 sound processor regularly, and provide a signed informed consent form.

The study included eight implanted subjects (three males and five females) aged between 43 and 72 years, with an average age of 61 years. Two patients had otosclerosis, two had sudden hearing loss, one had enlarged vestibular aqueduct syndrome, one had chronic otitis media, and two had a hearing loss of unknown origin.

In six cases, the CI was positioned on the right side, while only two patients had an implant on the left side. All subjects wore a HA in the contralateral ear (bimodal stimulation).

Subjects with cognitive impairment, or ocular disorders that prevented pupil dilation measurements, were excluded from the study. Cognitive impairment was evaluated by an experienced psychologist through the administration of cognitive screening tests (MoCA test).

Only 8 subjects were included in this study because, following the company’s voluntary recall, no more implants of that brand were performed in our hospital (https://www.salute.gov.it/imgs/C_17_AvvisiSicurezza_10493_azione_itemAzione0_files_itemFiles0_fileAzione.pdf, accessed on 16 February 2020).

### 2.3. Tested CI Microphones

All subjects were fitted with a Neuro 2 sound processor. This processor implements the Free Focus automatic adaptive multichannel directionality system. The sound processor can operate in three main dual-microphone directionality modes: omnidirectional (Omni mode), split-directional, and full-directional (Figure 1).

In the Omni mode, all directions in the room are equally represented, with sounds from the back gently attenuated and those from the front amplified. There are two Omni modes in the sound processor: Opti Omni and Speech Omni. Opti Omni is based on the difference between the front and rear microphones. Speech Omni enhances the Opti Omni strategy to better simulate the function of the pinna. Speech Omni is a light speech prioritization mode that enhances front focus and helps suppress sounds from the back. The split-directional (Split Dir) mode is an extended omnidirectional configuration. The response is omnidirectional in the low frequencies but in the high frequencies, starting at 2.000 Hz, the microphones have a higher sensitivity in the front direction. Split directionality is applied in moderately to noisy environments.

The full-directional mode favors the front of the microphone in all frequency bands. In this mode, sound coming from the front is prioritized, and background noise from the side or rear is filtered.

In the present study, Opti Omni, Speech Omni, and Split Dir were used. Full-directional was not analyzed as it is not included in the adaptive directional microphones.

### 2.4. Study Setting

Pupil recordings were made from all patients during a speech-in-noise test 24 months after the activation of the CI.

Participants were fitted with their Neuro 2 Sound Processor and the microphones were checked prior to testing. All tests were conducted using the Opti Omni, Speech Omni, and Split Dir modes of Free Focus.

The test sessions were conducted in a soundproof room under controlled lighting conditions [13].

The study included simultaneous speech audiometry and pupillometry measurements performed in the best aided condition (CI and contralateral HA).

The experiments were conducted in both quiet and noisy conditions. The speech signal and background noise were delivered through loudspeakers placed in a horizontal plane. Participants were seated in a stationary chair and kept at 100 cm from the loudspeakers at ear level.

The signal was presented at 0° azimuth, while the noise sources were positioned at 0°, +90°, and −90° azimuth, equally spaced. In the absence of noise, the signal came from the front (0° azimuth) (see Figure 2). The study evaluated the three noise conditions for all the microphones studied (Opti Omni, Speech Omni, and Split Dir), for a total of ten different conditions (one in quiet and nine in noise). As the CI side varied, the different configurations were named as follows: S0N0 (both signal and noise were delivered at 0° azimuth); S0Nic (signal came from the front, and noise was on the CI side); and S0Nctr (signal was at 0° azimuth, and noise was on the opposite side of the CI).

### 2.5. Materials

#### 2.5.1. Standard Hearing Tests

All patients underwent aided pure tone audiometry and speech audiometry in quiet conditions at 65 dBHL [14]. These tests were performed using a Madsen Astera audiometer (Natus Medical Incorporated, Denmark). The aided hearing threshold was measured as Pure Tone Average (PTA) (average hearing threshold across 500, 1000, 2000, and 4000 Hz).

Pure tone audiometry and speech audiometry in quiet conditions at 65 dBHL were carried out with the single CI and single HA.

#### 2.5.2. Carrier Speech-in-Noise Performance Test (HINT Test)

The experimental speech test was the Italian HINT sentences perception test.

Participants were trained to listen to a sentence and then to repeat it out loud.

The tests were carried out in both quiet and noisy conditions. During the quiet test, a list of 20 sentences was presented at an initial sound pressure level of 65 dBHL. The intensity level was then adjusted according to the participant’s responses. At the end of the list, the Speech Reception Threshold (SRT) was determined. The SRT is defined as the speech level corresponding to 50% of correct responses.

The noise trials used a HINT list and an adapted masker noise. The target speech signal was set at 65 dB SPL for all 20 sentences. The test initially started at +5 dB SNR, and after the first sentence, the noise level was adjusted based on the accuracy of repeating the sentence. Correct repetitions increased the masker level, while incorrect repetitions decreased it. The masker noise started three seconds before the sentence presentation and ended four seconds after the sentence offset. A speech-to-noise ratio (SNR) in dB was calculated at the end of the list.

#### 2.5.3. Pupillometry and Pupil Data Analysis

Pupil dilation was recorded during the HINT test. Participants were asked to focus on a dot in front of them and to inhibit eye blinking.

For this study, we used the Pupil Core eye tracking headset, a commercially available pupillometry device (Pupil Labs GmbH, Berlin, Germany; https://docs.pupil-labs.com/, accessed on 18 February 2020).

The analysis of pupil data was performed by evaluating the pupil curves. The traces obtained for all the sentences of a HINT list were averaged to obtain the mean curve and pupil dilation for each subject. The duration of the curves was estimated from 1 s before sentence onset to 4 s after sentence onset, which was the average noise offset. The raw data were resampled to 60 Hz to ensure a uniform sampling rate. Blinks were evaluated using the median absolute deviation [15], and data points between 35 milliseconds before and 100 milliseconds after a blink were discarded. Data points that were 2.5 standard deviations away from the mean, and less than 40 milliseconds in length, were also removed [16]. Data with more than 50% of the data points removed were excluded. Secondary filtering was applied to the data points using an average window of 50 ms. For each sentence, a baseline value was calculated as the mean pupil diameter measured during the 1 s period prior to the onset of the sentences while noise was presented. This value was subtracted from each curve to give a baseline corrected pupil curve.

This study focused on evaluating the peak pupil dilation (PPD) for each subject and condition. We extracted the PPD from the average curve, defining it as the maximum pupil dilatation within the time interval from sentence onset to the average noise offset.

#### 2.5.4. Cochlear Implant Fitting

Patients received appropriated adjustments to their CI sound processor prior to carrier speech testing. VoiceTrack was disabled and VoiceGuard was set to XDP medium.

CI fitting was performed (before the test) in the 1st, 3rd, 6th, and 12th month after CI activation.

### 2.6. Statistical Analysis

The effects of sound configuration and microphone type on the response variables (HINT and PPD) were examined using one-way and two-way Analysis of Variance (ANOVA) for repeated measures to account for within-subject result correlation, as detailed in Van Gog et al. [17].

One-way ANOVA was used to compare means when grouping observations by sound configuration (microphone type), irrespective of microphone type (sound configuration). Two-way ANOVA was used to assess their possible joint effects. An interaction between sound configurations and microphone type was also considered. Normality was assessed by testing the ANOVA residuals using Shapiro–Wilk tests. Pairwise comparisons of means were carried out using Games-Howell post hoc tests [18].

Whenever the *p*-value was not directly reported, a test was referred to as statistically significant if the associated *p*-value was below α = 0.05. When conducting post hoc comparison tests, this threshold was lowered to account for multiplicity, adopting a Bonferroni correction.

The relationship between HINT and PPD was examined by means of Spearman rank correlation, as the normality assumption failed for the SNR variable.

All calculations were carried out using R Statistical Software [19]. Among the many alternative R packages that can be used for this task, we consider nlme [20], as the most frequently used [21,22].

## 3. Results

The hearing tests performed showed that the mean aided PTA with the CI was 42.7 dB HL (range 35–52), whereas the aided PTA with the contralateral HA was 50.5 dB HL (range 29–75). Speech discrimination in quiet conditions at 65 dB SPL was 79.2% with the CI (range 60–100) and 65.6% with the HA alone (range 15–100).

Table 1 shows the mean and standard deviation values for SNR (in the HINT test) and pupillometry (PPD) across different noise configurations and using different types of microphones. Our results show that when groups are formed on the basis of sound configurations, the one-way ANOVA demonstrates that the differences between S0N0 and S0Nctr and S0Nic are significant (*p* value < 0.01). In particular, the SNR under S0N0 differs significantly from the other two configurations, whereas the difference between the S0Nctr and S0Nci groups is not statistically significant.

Although the mean SNR values are not significantly correlated with the groups defined by the three microphones (*p* > 0.05), a higher value can be observed when the Opti omnidirectional microphone is used. There is a high standard deviation in all configurations. The ANOVA null hypothesis is not rejected in the analysis of the PPD either between the sound configurations or between the types of microphones used.

On closer inspection, using a two-way repeated measures ANOVA with an interaction model, the effect of sound configurations on SNR remains statistically significant (*p*-value < 0.01). However, the interaction between microphone type used and sound configuration, shown in Table 2, is not statistically significant.

Figure 3 shows the average PPD using different microphones for the sound configuration. It is evident that there is no difference in PPD between quiet (S0) and noise configurations (S0N0, S0Ncrt, and S0Nci). A greater pupil dilation was observed with a speech omnidirectional microphone in the S0Nctr condition. In contrast, higher PPD values were observed with a split-directional microphone for the S0N0 and S0Nci configurations. Both the split-directional and Opti omnidirectional microphones produce similar PPD values when the noise is positioned to the side of the cochlear implant (S0Nci).

Pupillometry curves were examined for all three sound configurations, as shown in Figure 4. A similar trend of increased pupil dilation from baseline is observed in all sound configurations from 3 s after the start of the test. The curves show a decrease from 6 s to the end of the test. The time from 0 to 4 s corresponded to listening to the sentence, the time from 4 to 6 s was allotted for repetition, and after 6 s, the time to return to baseline was given.

In the S0N0 configuration, it can be observed that the Split Dir curve shows a greater pupil dilation during the repetition time (4–6 s) compared to the overlapping Speech Omni and Opti Omni tracks. On the other hand, the pupil dilation curve is lower in the S0Nci configuration with the speech omnidirectional microphone. When the noise was placed on the contralateral side of the cochlear implant (S0Nctr), no differences were found in the curves with the three microphones. It is important to note that all measurements have a large standard deviation, as indicated by a cloud around each line.

### Correlation between SNR and PPD

We investigated whether there was a correlation between SNR and peak pupil dilation to determine if a higher SNR was associated with increased pupil dilation. We did not find a statistically significant correlation between the two parameters (r = 0.064, *p* value = 0.595; see also Figure 5).

## 4. Discussion

The aim of this study was to evaluate the effectiveness of adaptive microphones in reducing listening effort and to investigate the relationship between listening ability in noise and listening effort in a sample of CI recipients.

Three types of microphone and three different sound configurations were evaluated in this study and pupillometry was used to objectively assess listening effort.

Our results showed that there is no effect of different microphones on the listening effort required for different sound configurations.

These data may be related to the type of bimodality of the subjects in the study.

The effectiveness of CI adaptive microphones in reducing effort were evaluated under ‘ecological’ listening conditions. Specifically, the acoustic amplification and microphone filters of the HA were not modified. Conversely, different types of microphones were evaluated with the CI, and, consequently, the two devices did not provide true bimodal compensation.

In addition, bimodal stimulation results in different sound processing between the two ears. People with normal hearing have a natural ability to spatially separate sound sources. This phenomenon is known as “Spatial Release from Masking” (SRM) and is the basis for improved speech perception in noise. The basic mechanisms involved in SRM are the head shadow, the squelch, and the binaural summation effects [23]. These mechanisms are based on the processing of interaural time differences (ITDs) and interaural loudness differences (ILDs).

In bimodal listeners, as the subjects in this study, the benefit may be influenced by differences in ITD processing between the CI and the HA. Normally, the CI delivers auditory information with shorter processing latencies than the HA.

The ILD processing of our patients was probably also deficient because of the interaural misalignment of the aided thresholds and the different preprocessing of the signal at different loudness levels between the ears.

For the reasons listed above, differences in HA filters and in ITD and ILD processing may be the cause of the non-effects of different microphones.

Our data show, on the contrary, that sound configurations are factors that influence the listening capacities of these patients. We found that the SNR reported in the HINT test differed depending on whether the noise was presented in the same position as the signal or whether the two sources were in different positions. When the signal and noise were presented at 0°, subjects required on average a lower SNR than in S0Nci or S0Nctr configurations. These results showed that subjects required a higher ratio to hear sentences when speech and noise came from different directions.

As reported above, with the patients in this study being bimodal, ITD and ILD processing are modified and, consequently, the SRM and the speech perception in noise abilities may also be modified. In accordance, the literature suggests that bimodal users can benefit from different sound source separation, but SRM is generally poorer compared to normal hearing people due to the limited squelch effect and binaural summation [24]. On the contrary, in this study, the speech and noise originate from the same frontal source (S0N0), the ITD and ILD processing are only marginally used, hearing depends mainly on the binaural summation effect, and speech perception in noise gets better.

However, as reported in the literature, other factors can affect listening in different sound configurations, such as the orientation of the sound sources [25], the type of noise used [24], and the typology of the CI microphones [26,27,28].

Pupillometry data do not show strong correlations with microphone types and sound configurations either.

In our data, the global morphology of the pupil curves is similar in all configurations. We found an intrinsic difference in all the pupil curves with a higher pupil dilation during the time of repetition of the sentences compared to the initial listening part. These differences indicate a higher listening effort during the repetition phase compared to the listening phase, because two different capacities are involved. In fact, the repetition part requires a working memory capacity, which requires more attention to the task, resulting in more listening effort. However, considering the quantitative parameter of the curve, no significant differences were observed between the different listening configurations and the different CI microphones. PPD was therefore not correlated with the SNR, which was statistically lower (better) in the S0N0 configuration.

We found only a trend difference in the HINT test between directional (Opti Omni) and spatial (Split Dir) CI microphones: a higher SNR is required for the directional microphone. For these reasons, we are investigating whether the typology of the environment could affect hearing performances and pupillometry. In the case of S0N0, a slight increase in PPD was found with the Split Dir microphone. The functionality of this type of microphone is based on a dual sensitivity for low and high frequencies, with a preference for speech cues coming from the front. The pupillometry measurements therefore reflect these characteristics.

In the case of noise presented on to the CI side (S0Nci), we found a higher similar PPD with both the Split Dir and the Opti omnidirectional microphones. The lower PPD with the Speech Omni microphone is due to the intrinsic mechanism which involves a filtering of the background noise presented near the CI and a higher focus on the sound source in front of the subject. This involved a reduction in effort and in PPD. Higher pupil dilation and listening effort were found with the other two microphones. Opti Omni microphones have less filtering than speech omnidirectional ones and pick-up sounds in all directions.

When the noise was presented on the HA side (S0Nctr), no major differences were found between the three microphone curves, but an average higher PPD was shown with the speech omnidirectional microphone. In this configuration, the noise source is contralateral to the CI. The hearing device, regardless of the microphone used, applies an attenuation mechanism that simulates the head shadow effect, but does not use the filtering mode. The slight increase in pupil dilation with the speech omnidirectional microphone reflects a higher listening effort compared to the other two, not only due to the attenuation mechanism, but also to the higher focus on the speech source.

It is possible that we did not find large differences in PPD in the different conditions because the task was the same in all situations. This suggests that CI users in our study were allocated similar listening effort when processing different noise and microphone situations.

A limiting factor in this study is the predominance of elderly population (mean age of 61 years). In fact, the elderly show, in general, a poorer level of attention and less accuracy in the location of the sound source than younger people [29,30]. Furthermore, the pupillometry test can also be influenced. The literature reports that elderly subjects or subjects with encephalic pathologies show different pupillometric traces compared to younger subjects [31,32].

## 5. Conclusions

In conclusion, the different types of microphones offered by directional microphones of Neuro CI had different effects on the listening of bimodal patients. A slight correlation was found between the use of omnidirectional or directional beamformer microphones and sound source orientation. However, the difference between the two-side devices used, the orientation of the sound source, and the type of hearing test used are factors that have a significant impact on the listening effort results. However, the pupillometry measurements do not significantly correlate with the different microphone types.

In the future, it will be necessary to study pupil dilation with a constant SNR and controlled device characteristics in order to better understand the real correlation between microphone type, sound source, and listening effort. It will also be interesting to investigate the listening effort abilities over time in order to assess whether the long-term use of the CI is an impacting factor.

## Figures and Tables

**Figure 1 jcm-13-01134-f001:**
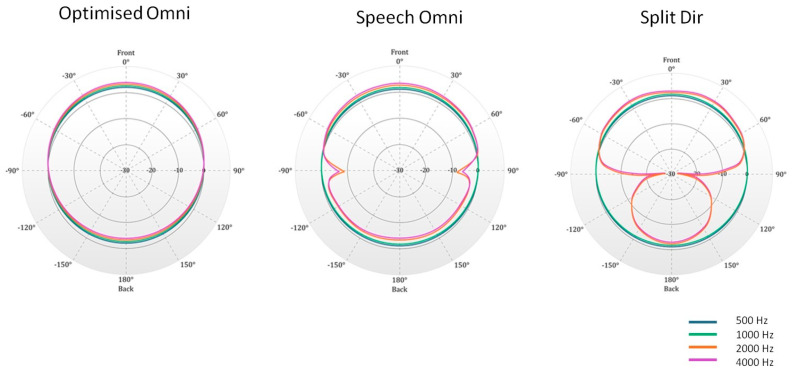
The three types of microphones used in this study. The blue line indicates 500 Hz, the green line 1000 Hz, and the red line 2000 Hz.

**Figure 2 jcm-13-01134-f002:**
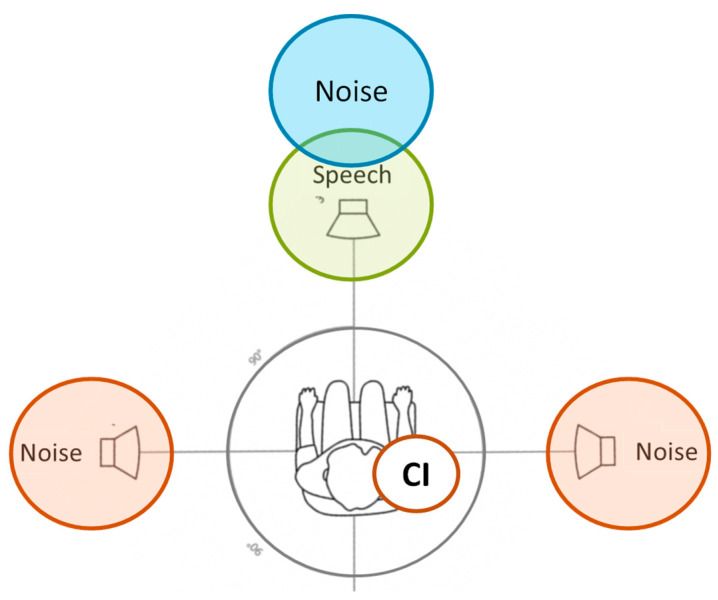
Experimental setup. The loudspeakers were positioned horizontally at ear level. The speech was presented at 0° azimuth, while the noise sources were placed at 0°, +90°, and −90° angles. The distance between the sound sources and the listener’s head center was 100 cm.

**Figure 3 jcm-13-01134-f003:**
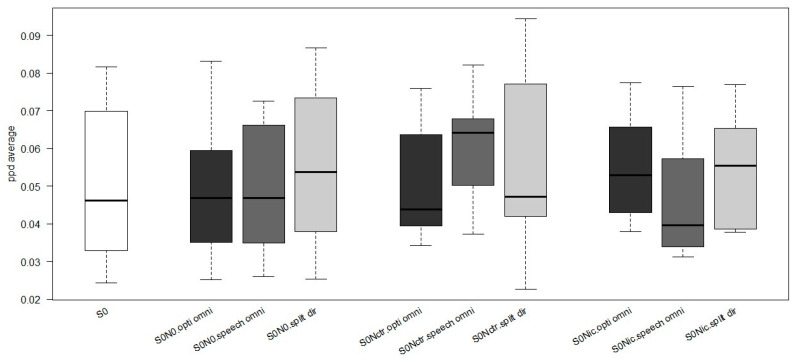
Normalized mean peak pupil dilation (PPD) in different sound configurations using different microphones. In with the quiet condition (S0), in black Opti Omni, in dark grey Speech Omni and in light grey Split Dir conditions.

**Figure 4 jcm-13-01134-f004:**
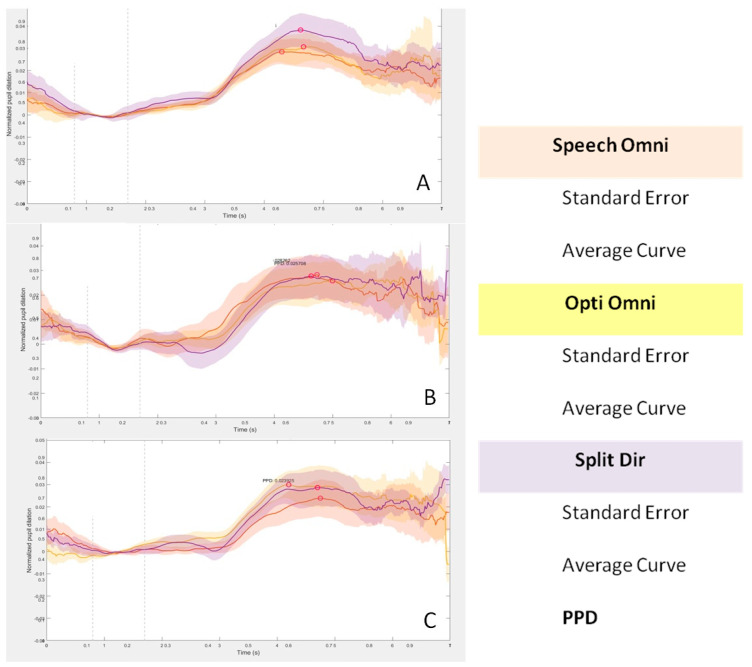
Pupillometric curves in three sound conditions with S0N0 in box (**A**), S0Nctr in box (**B**), and S0Nci in box (**C**). Data from the Speech Omni microphone are shown in orange, the Opti Omni microphone in yellow, and the Split Dir microphone in purple.

**Figure 5 jcm-13-01134-f005:**
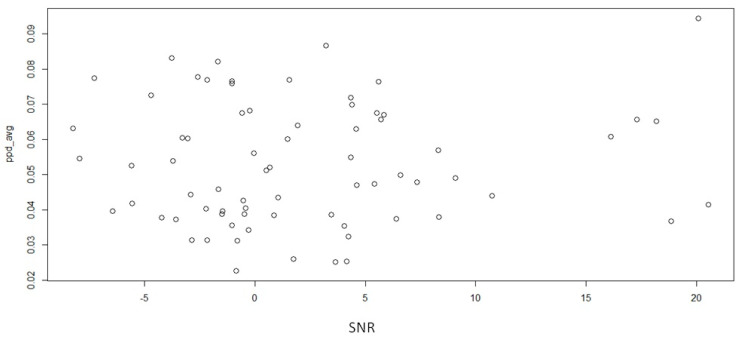
SNR and PPD average correlation.

**Table 1 jcm-13-01134-t001:** Speech-to-noise ratio (SNR) in the HINT test and peak pupil dilation (PPD) in pupillometry for different noise configurations (S0N0, S0Nctr, and S0Nci) and different microphones (Speech Omni, Opti Omni, Split Dir).

	SNR (in dB)		Peak Pupil Dilation (PPD) (in au)	
Sound Configuration	Mean	Standard Deviation	Mean	Standard Deviation
S0N0	−0.11	3.09	0.0512	0.0188
S0Nctr	+3.13	7.67	0.0557	0.0180
S0Nci	+3.46	7.36	0.0518	0.0149
Microphone				
Speech Omni	+1.43	6.49	0.0520	0.0166
Opti Omni	+3.15	6.87	0.0514	0.0156
Split Dir	+1.89	6.30	0.0552	0.0196

**Table 2 jcm-13-01134-t002:** An interaction model is presented for SNR evaluation, which examines the correlation between different sound configurations and microphones.

	Num (DF)	Den (DF)	F-Value	*p*-Value
(Intercept)	1	56	1.376212	0.2457
Sound configuration	2	56	5.064806	0.0095
Microphone	2	56	1.024253	0.3657
Configuration/microphone	4	56	0.623330	0.6478

## Data Availability

The data presented in this study are available on request from the corresponding author. The data are not publicly available because they are sensitive health data.

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
