# Peer review of "Do Different Types of Microphones Affect Listening Effort in Cochlear Implant Recipients? A Pupillometry Study"

_jcm, 2024, doi:10.3390/jcm13041134_

Round 1

Reviewer 1 Report

Comments and Suggestions for Authors

Overall: The English can be cleaned up, a lot of grammatical issues. Please have a native English speaker read through the whole document.

The abstract:

* you need to actually report results, not just say significant or not. You need to give much more information. 

* I even wonder if you report not significant correlations in an abstract

Introduction

* You use the same abbreviation for cochlear implant and cochlear implantation. Make up your mind which one you want. 

* line 57: I have no clue what you are trying to say here

* line 59: not speech in noise BUT speech PERCEPTION in noise

* line 62: you have used the abbreviation of cochlear implant so use it here

MATERIALS AND METHODS

* have your patients all signed an informed consent? If not that could be a violation of ethics

* line 73-74: please rewrite this sentence

* line 77: what do you mean with consecutively?

* line 80: how did you know they didn't have mental health issues?

* do you know any frequency configuration about the microphones used? You copy-pasted nicely the marketing info from Oticon Medical but for this kind of research you need more detailed info on the microphones

* line 138: are you sure the speech was presented at 65dBSPL? Nowadays, most audiometers are calibrated using a dBHL scale.

* I am guessing your α value is 0.05 (and less if you have to adjust for Type I error) but you don't say.

* you say you are using the Spearman correlation, so I am guessing that was because results were not normally distributed. You don't say why you picked that test.  

* line 187: the what???

RESULTS

* Line 189: only here you say you have 8 subjects. That info should have been part of your methods.

* first and second paragraph: that info doesn't belong here but in the methods.

* line 198: all of a sudden you start talking about HA without explaining what that stands for. 

* 3rd paragraph: some of the info here doesn't belong here but in methods. And that last sentence: I have no clue what you are trying to say here.  

* Line 210: Your results section just just be a dry report about your data. If you start talking about rejecting your hypothesis (which you never formulated by the way) that goes into your discussion.

* Table 1: what is the unit for pupillometry?

* line 251: if you have abbreviated speech-to-noise ratio once, than stick with it.

* line 253: your reporting of your results is not according to that standard of how the should be reported

DISCUSSION

* Overall, the discussion is written in such a way that it's difficult to follow.

* You don't say the results are difficult to explain. You explain, end of it. You say that your results are different from reports in the literature. I don't see any of that.

* The abbreviations are all over the place, sometimes they are used, sometimes not. Very confusing.

CONCLUSIONS

* Fee Focus Light directionality. Where is that concept coming from, all of a sudden? 

* line 311: Using a Latin Square to randomize your tests and lists could help. I don't see anything about that.

Comments on the Quality of English Language

The English can be cleaned up, a lot of grammatical issues. Please have a native English speaker read through the whole document.

Author Response

Reviewer 1 Comments

Overall: The English can be cleaned up, a lot of grammatical issues. Please have a native English speaker read through the whole document.

According to the Editor’s suggestion, a native English speaking person has performed paper copy-editing.

The abstract:

 you need to actually report results, not just say significant or not. You need to give much more information.

I even wonder if you report not significant correlations in an abstract

In according with the reviewer, we modified results and conclusion paragraphs in the abstract.

Introduction

You use the same abbreviation for cochlear implant and cochlear implantation. Make up your mind which one you want.

We modified the CI abbreviations. We used CI for Cochlear Implant throughout the paper

line 57: I have no clue what you are trying to say here

We remove the part of the sentence that cannot be understood and we modified the sentence in  the following: “The interest of pupillometry is that it can be applied for the development of new clinically relevant objective measures of speech in noise in CI users”

* line 59: not speech in noise BUT speech PERCEPTION in noise

We modified “speech in noise” with “speech perception in noise”

* line 62: you have used the abbreviation of cochlear implant so use it here

As suggested, we used CI abbreviation

MATERIALS AND METHODS

* have your patients all signed an informed consent? If not that could be a violation of ethics

In accordance with the rules of the Ethics Committee, all patients signed an informed consent form before the start of the study. This data is reported in the Participants paragraph. We added also the following sentence in the Study design paragraph:” All subjects signed the Patient Informed Consent Form before the first assessment”.

* line 73-74: please rewrite this sentence

We rewrite the sentence in the following: “The participants in this study are bimodal patients and they use a CI in the worst hearing side and a hearing aid (HA) in the controlateral side. Everyone had been using the CI for 24 months”

* line 77: what do you mean with consecutively?

We remove “consecutively”

* line 80: how did you know they didn't have mental health issues?

We screened the cognitive abilities with MoCA test. In agreement with the review, we have removed the we statement concerning the mental health because it was not evaluated by validated test. We modified the sentence in the following: “Subjects with cognitive impairment, or ocular disorders that prevented pupil dila-tion measurements were excluded from the study. Cognitive impairment was evaluated by experienced psychologist through the administration of cognitive screening tests (MoCA test).”

* do you know any frequency configuration about the microphones used? You copy-pasted nicely the marketing info from Oticon Medical but for this kind of research you need more detailed info on the microphones

We added the following sentences: “Speech omni is a light speech prioritisation mode that has enhanced front focus and helps suppress sounds from the back.” and “Split directionality is applied in moderately to noisy environments.”. We add in the figure 1 caption the frequencies of the three microphones in our study: “Blue line indicates 500 Hz, green line 1000 Hz and red line 2000 Hz.”

* line 138: are you sure the speech was presented at 65dBSPL? Nowadays, most audiometers are calibrated using a dBHL scale.

The speech was presented at 65 dB HL. We modified in the text the unit of measurement

* I am guessing your α value is 0.05 (and less if you have to adjust for Type I error) but you don't say.

* you say you are using the Spearman correlation, so I am guessing that was because results were not normally distributed. You don't say why you picked that test. 

We modified the sentences in the following: “Whenever the p-value is not directly reported, a test is referred to as statistically significant if the associated p-value is below alpha=0.05. When conducting post-hoc comparison tests this threshold is lowered to account for multiplicity adopting a Bonferroni correction.”

The relationship between HINT and PPD is examined by means of Spearman rank correlation, as the normality assumption fails for the SNR variable.

* line 187: the what???

We modified the sentence in the following: “All calculations were carried out using R Statistical Software [19]. Among the many alternative R packages that can be used for this task, we consider nlme [20], as the most frequently used”

RESULTS

* Line 189: only here you say you have 8 subjects. That info should have been part of your methods.

We have moved this information concerning the study participants to the Methods section.

* first and second paragraph: that info doesn't belong here but in the methods.

We have moved this information concerning the study participants to the Methods section.

* line 198: all of a sudden you start talking about HA without explaining what that stands for.

Use of hearing aids is  reported in Materials and methods paragraph. In particular in “Study design” and “Participants” paragraphs is reported that all patients had a CI in a side and a HA in contralateral side. We added the following sentence in the “3.5. Materials- 3.5.1. Standard Hearing tests” paragraph: “Pure tone audiometry and speech audiometry in quiet at 65 dBHL were carried out with the single CI and single HA.”

* 3rd paragraph: some of the info here doesn't belong here but in methods. And that last sentence: I have no clue what you are trying to say here. 

We added the following sentence in the “3.5. Materials- 3.5.1. Standard Hearing tests” paragraph: “Pure tone audiometry and speech audiometry in quiet at 65 dBHL were carried out with the single CI and single HA.”

* Line 210: Your results section just be a dry report about your data. If you start talking about rejecting your hypothesis (which you never formulated by the way) that goes into your discussion.

We modified the sentence in the following: “Our results show that when groups are formed on the basis of sound configurations, the one-way ANOVA demonstrate that the differences between S0N0 and S0Nctr and S0Nic are significant (p value <0.01).”

* Table 1: what is the unit for pupillometry?

The unit for pupillometry is arbitrary units (au). We added the unit in table 1.

* line 251: if you have abbreviated speech-to-noise ratio once, than stick with it.

We modified “speech-to-noise ratio” with the abbreviation SNR.

* line 253: your reporting of your results is not according to that standard of how the should be reported

We modified the sentence in the following:” "We did not find a statistically significant correlation between the two parameters (r=0.064, p value = 0.595; see also Figure 5)"

DISCUSSION

* Overall, the discussion is written in such a way that it's difficult to follow.

We modified the discussion paragraph

* You don't say the results are difficult to explain. You explain, end of it. You say that your results are different from reports in the literature. I don't see any of that.

We removed the sentences “These results are somewhat unexpected and difficult to explain. We will try to explore these results in the following paragraphs” and we added the following sentence: “These results can be explained by the type of test used.” and “Pupillometry data is also difficult to interpret”.

* The abbreviations are all over the place, sometimes they are used, sometimes not. Very confusing.

We have uniformed abbreviations.

CONCLUSIONS

* Fee Focus Light directionality. Where is that concept coming from, all of a sudden?

We modified with “directional microphones of Neuro CI”

* line 311: Using a Latin Square to randomize your tests and lists could help. I don't see anything about that.

We think that Latin Square is not applicable in our study because is not a study over time, it is not a randomize study and the same treatment (in our case pupillometry) is not assigned to each time period the same number of times and to each subject the same number of times (such as Latin square definition)

Comments on the Quality of English Language

The English can be cleaned up, a lot of grammatical issues. Please have a native English speaker read through the whole document.

According to the Editor’s suggestion, a native English speaking person has performed paper copy-editing

Reviewer 2 Report

Comments and Suggestions for Authors

The authors present an interesting topic concerning patients with hearing implants, where their results are consistent with observations from daily clinical work. It is noteworthy that the paper is based on observations of only 8 patients (why such a small group?), in addition to a predominantly elderly population, which may suggest some limitations of the paper. It can be expected that the elderly may present a poorer level of attention and less attention to interpreting the location of the sound source than younger people. In the literature, only 8 items out of 26 are from the last 5 years. It would be worthwhile to think about extending the study with a longer follow up and a step-by-step observation, i.e., how the observed parameters change after one, two and more years. There is also a lack of information on how conducted (how often were the speech processor settings) in the analyzed patients.

Author Response

The authors present an interesting topic concerning patients with hearing implants, where their results are consistent with observations from daily clinical work.

Thank you for the encouraging comments.

It is noteworthy that the paper is based on observations of only 8 patients (why such a small group?),

In according with the reviewer we added the following sentence in the Materials and Methods-Participants paragraph: “Only 8 subjects were included in this study because, following the company's voluntary recall, no more implants of that brand were performed in our hospital (https://www.salute.gov.it/imgs/C_17_AvvisiSicurezza_10493_azione_itemAzione0_files_itemFiles0_fileAzione.pdf).”

In addition to a predominantly elderly population, which may suggest some limitations of the paper. It can be expected that the elderly may present a poorer level of attention and less attention to interpreting the location of the sound source than younger people.

In according to the review, we added the following sentences in the discussion paragraph: “A limiting factor in this study is the predominance of elderly population (mean age of 61 years). In fact, elderly show, in general, poorer level of attention and less accuracy in location of the sound source than younger people (Pupo et al 2022, Dai et al 2022). Furthermore, pupillometry test can also be influenced. Literature reports that elderly subjects or subjects with encephalic pathologies show different pupillometric traces compared to younger subjects (Lenga et al 2023, Zekveld et al 2011).”

In the literature, only 8 items out of 26 are from the last 5 years.

We have added more recent articles:

Mundo, A. I., Tipton, J. R., & Muldoon, T. J. (2022). Generalized additive models to analyze nonlinear trends in biomedical lon-gitudinal data using R: Beyond repeated measures ANOVA and linear mixed models. Statistics in Medicine, 41(21), 4266-4283.

Pupo DA, Small BJ, Deal JA, Armstrong NM, Simonsick EM, Resnick SM, Lin FR, Ferrucci L, Tian Q. J Hearing and Mobility in Aging-The Moderating Role of Neuropsychological Function. Gerontol A Biol Sci Med Sci. 2022 Oct 6;77(10):2141-2146. doi: 10.1093/gerona/glac047.

Dai L, Best V, Shinn-Cunningham BG. Sensorineural hearing loss degrades behavioral and physiological measures of human spatial selective auditory attention. Proc Natl Acad Sci U S A. 2018 Apr 3;115(14):E3286-E3295. doi: 10.1073/pnas.1721226115.

Lenga P, Kühlwein D, Schönenberger S, Neumann JO, Unterberg AW, Beynon C. The use of quantitative pupillometry in brain death determination: preliminary findings. Neurol Sci. 2023 Dec 12. doi: 10.1007/s10072-023-07251-4.

It would be worthwhile to think about extending the study with a longer follow up and a step-by-step observation, i.e., how the observed parameters change after one, two and more years.

In according with the review, we have added the following sentence in Conclusion paragraph: “It will also be interesting to investigate the listening effort abilities over time in order to assess whether long-term use of the CI is an impacting factor.”

There is also a lack of information on how conducted (how often were the speech processor settings) in the analyzed patients.

In according with the review, we have added the following sentence in Cochlear Implant Fitting paragraph “CI fitting was performed (before the test) at the first, 3th, 6th, 12th month after CI activation.”